# Assessment of the Versius Robotic Surgical System in Minimal Access Surgery: A Systematic Review

**DOI:** 10.3390/jcm11133754

**Published:** 2022-06-28

**Authors:** Ibrahim Alkatout, Hamid Salehiniya, Leila Allahqoli

**Affiliations:** 1Kiel School of Gynaecological Endoscopy, Campus Kiel, University Hospitals Schleswig-Holstein, Arnold-Heller-Str. 3, Haus 24, 24105 Kiel, Germany; 2Social Determinants of Health Research Center, Birjand University of Medical Sciences, Birjand 9717853577, Iran; alesaleh70@yahoo.com; 3Ministry of Health and Medical Education, Tehran 1467664961, Iran; lallahqoli@gmail.com

**Keywords:** versius surgical robot system, new robotic platform, visceral surgery, general surgery, gynecology surgery, urologic surgery, minimal access surgery

## Abstract

Background: Despite the superiority of minimal access surgery (MAS) over open surgery, MAS is difficult to perform and has a demanding learning curve. Robot-assisted surgery is an advanced form of MAS. The Versius^®^ surgical robot system was developed with the aim of overcoming some of the challenges associated with existing surgical robots. The present study was designed to investigate the feasibility, clinical safety, and effectiveness of the Versius system in MAS. Materials and Methods: A comprehensive search was carried out in the Medline, Web of Science Core Collection (Indexes = SCI-EXPANDED, SSCI, A & HCI Timespan), and Scopus databases for articles published until February 2022. The keywords used were Versius robot, visceral, colorectal, gynecology, and urologic surgeries. Articles on the use of the Versius robot in minimal access surgery (MAS) were included in the review. Results: Seventeen articles were reviewed for the study. The investigation comprised a total of 328 patients who had been operated on with this robot system, of which 48.3%, 14.2%, and 37.5% underwent colorectal, visceral, and gynecological procedures, respectively. Postoperative and major complications within 30 days varied from 7.4% to 39%. No major complications and no readmissions or reoperations were reported in visceral and gynecological surgeries. Readmission and reoperation rates in colorectal surgeries were 0–9%. Some procedures required conversion to conventional laparoscopic surgery (CLS) or open surgery, and all procedures were completed successfully. Based on the studies reviewed in the present report, we conclude that the Versius robot can be used safely and effectively in MAS. Conclusions: A review of the published literature revealed that the Versius system is safe and effective in minimal access surgery. However, the data should be viewed with caution until randomized controlled trials (RCTs) have been performed. Studies on the use of this robotic system in oncological surgery must include survival as one of the addressed outcomes.

## 1. Introduction

Robotic-assisted procedures revolutionized minimal access surgery (MAS) and overcame the technical limitations of laparoscopy [1,2]. With robotic assistance, the indications for MAS could be extended to include delicate and complex procedures. The advantages included three-dimensional (3D) magnification, high-resolution (HD) visualization, greater dexterity, tremor filtration, and extreme precision of movement [2,3]. However, the current generation of robots have certain disadvantages: apart from the bulkiness of the robot platforms, plastic biomedical waste from disposable trocar and robotic instrument use leads to the generation of tons of CO_2_ emissions that may have a significant impact on the environment [4,5,6]. This increased environmental impact of robotic-assisted surgery may not sufficiently offset the clinical benefit [4,5].

After FDA approval, robot-assisted surgery was first used in 2000. A wide range of robot-assisted surgery devices have been developed since, including the Automated Endoscopic System for Optimal Positioning (Computer Motion, Santa Barbara, CA, USA), the Zeus Surgical System (Computer Motion), and the da Vinci Surgical System (DVSS; Intuitive Surgical Inc., Mountain View, Sunnyvale, CA, USA) [7,8,9,10].

Over the last twenty years, the da Vinci device emerged as the predominant system in the robot-assisted surgery market. It is a master-slave tele-manipulation system that provides high-resolution 3D images. However, the system permits the surgeon to perform endoscopic surgery only if the ports are positioned appropriately and no arm collides with other arms [11]. The recent expiry of the patents of Intuitive Surgical Inc.—manufacturers of the da Vinci robot—has allowed new systems to enter the market [12]. Competitors of the da Vinci system have been launched in the robotic market [13,14]. Ongoing innovations in robotic technology have led to new features such as the open-console design, haptic feedback, smaller instruments, greater ease of movement, low costs, greater flexibility of use, and separately mounted robotic arms [9,12,13,15,16]. Although the purpose of the modifications was to improve the existing technology, reports suggest that shifting from one model to another poses certain challenges for surgeons [17,18,19].

The Versius surgical robotic system was recently launched by Cambridge Medical Robotics (CMR) [20,21]. The ergonomic platform is equipped with an open console that permits the user to operate the device in a standing or sitting position, thus reducing stress and fatigue (Figure 1A,B). The surgeon may use up to five lightweight robotic arms, each existing as a solitary robotic unit for greater freedom of port placement. V-wrist technology permits 360 degrees of wrist motion, 7 DOF, and haptic feedback [22]. According to Puntambekar and coworkers who evaluated the feasibility of the Versius system, its main advantage is the presence of individual robotic arms that actually mimic the laparoscopic arms. Since the robotic and laparoscopic port placements are identical, the device permits duplication of laparoscopic steps [20].

Characteristics of the Versius robotic system are summarized in Table 1.

A year after the Versius surgical robotic system was introduced in the market, many surgeons started to use it in visceral, gynecological, and urological surgery despite a limited body of scientific data concerning its feasibility and safety [21]. In conjunction with the primary use of the system in MAS, scientists performed preliminary studies in human cadavers and live animals as well as preclinical investigations to evaluate the operational safety and feasibility of the system [13,20,22,23,24,25,26,27,28,29,30,31,32,33,34,35]. Given the novelty of the device and the small number of existing studies, scientists and surgeons have been unable to draw final conclusions about the safety and effectiveness of the robot. A systematic review of the safety and usability of the robot will be essential to establish the surgeon’s confidence in using the device for MAS.

Medical advances based on the integration of artificial intelligence, machine learning, and augmented realities are widespread and have benefited a large number of patients [36]. The advent of the robotic era has served as an incentive for new systems and technologies aimed at enhancing patient care. The purpose of the current systematic review was to determine the feasibility, clinical safety, and effectiveness of the Versius system in MAS.

## 2. Materials and Methods

The present systematic review was conducted over a period of four months (November–February 2022). Its aims were to determine:(a)the feasibility of the Versius system for healthcare professionals;(b)the clinical safety and effectiveness of the Versius system in MAS.

The concept of the Versius system was developed by Luke Hares to address a number of identified needs of surgeons. The needs were confirmed by discussions with surgeons, and had not been resolved by the existing surgical robots [13]. External views and endoscopic views of the Versius system are shown in Figure 2 and Figure 3.

### 2.1. Search Strategy and Sources of Information

The study was conducted in accordance with the PRISMA (preferred reporting items for systematic reviews and meta-analyses) and AMSTAR (assessing the methodological quality of systematic reviews) guidelines [37]. Three databases—PubMed/MEDLINE, Scopus, and Web of Science—were searched for relevant articles. The search was performed in November and February 2022 using the following keywords: “Versius robot”, “new robotic platform”, “robot-assisted surgery”, “gynecology”, “urology”, “intestine”, “hysterectomy”, “myomectomy”, “renal”, “prostatectomy”, “ureteral”, “visceral”, “colon”, “rectal”, “tumor”, “abdominal surgery”, “surgeon training”, “cholecystectomy”, “nephrectomy”, “minimal surgery”, “colorectal”, “general procedures”, “oncology”, “live animals”, “live humans”, “human cadavers”, and “ learning curve”. Boolean (AND, OR) operators and Medical Subject Headings (MeSH) terms were then used to optimize the selection of records. In order to obtain an even more comprehensive body of data, we performed a manual search of reputable journals as well as a manual search of references in full-text articles and related systematic reviews.

### 2.2. Inclusion and Exclusion Criteria

We reviewed peer-reviewed studies conducted throughout the world and considered all types of investigations. Studies in the English, German, French, and Italian languages were taken into account. Our research revealed no published articles in languages other than English. Studies published in the English language, using the Versius system in colorectal, visceral, and gynecological surgery were reviewed. No restrictions were imposed in terms of time or study type.

### 2.3. Study Selection

The review process consisted of two screening steps: (a) reviewing the title and summary of the articles, and (b) reviewing the full text of the articles. For the first level of screening, the titles and abstracts of the articles were read and analyzed independently by two researchers (IA) to identify eligible articles. In the second step, two researchers (IA, HS) evaluated the full text of each article independently. All retrieved articles were entered into a database on Endnote X7.

### 2.4. Quality of the Articles

Depending on the type of investigation, two scales were used to assess the quality of the studies:

(a) The National Institutes of Health (NIH) quality assessment tool for case series studies, last updated in July 2021 [38], consists of nine questions and assigns three quality ratings (good, fair, and poor).

(b) The methodological index for non-randomized studies (MINORS) tool includes eight questions and divides articles into two qualitative ranks (low and high risk of bias) [39].

### 2.5. Data Extraction

Two independent investigators extracted pertinent data from the studies, including the authors, the year of investigation, country, study design, sample size, audience group, length of program, type of surgery, and main results.

## 3. Results

### 3.1. Search Results

Eighty-three articles were found in the databases, of which seventeen met the inclusion criteria (Figure 3).

Seventeen studies published between 2019 and 2022 were deemed eligible for the review. Based on the methodology, the following types of studies were included: pilot studies (*n* = 6), clinical trials (*n* = 3), case series (*n* = 3), observational studies (*n* = 1), and unknown (*n* = 4). Based on the quality tools we used, one of the experimental studies had a low risk of bias and seven had a high risk of bias due to small sample sizes. In the case series studies, six were of good quality and two of fair quality. The investigations had been conducted in Germany (*n* = 1), the United States (USA) (*n* = 3), the United Kingdom (UK) (*n* = 8), the USA and the UK (*n* = 1), and India (*n* = 4). The investigations had been performed on cardboard boxes (*n* = 1) (40); human cadavers (*n* = 5) [13,22,23,27,28]; human cadavers and live animals (*n* = 3) [24,25,26]; and live humans (*n* = 7) [20,29,30,31,32,33,34,35]. Results were divided into the following two categories: (a) usability, safety, and effectiveness of the Versius system (9 studies); and (b) clinical safety and effectiveness of the Versius system for use in MAS (8 studies). Characteristics of the studies included in the review are summarized in Table 2.

### 3.2. Feasibility of the Versius Robotic System

The Versius system was developed using feedback from end-users throughout the design process, and aimed to minimize barriers to the uptake of robotic MAS [13]. The usability, safety, and effectiveness of the Versius system were assessed in the first published studies addressing the application of this robot [13,22,23,25,26,28,40]. To date, 108 procedures (8 cardboard boxes, 83 in human cadavers, and 17 in live animals) have been performed by 75 surgical teams in colorectal and visceral surgery, gynecology, urology, and the nasopharynx. Fifty-two of the procedures were performed in general and colorectal surgery, and twenty-three in gynecology. Radical nephrectomy, prostatectomy, and pelvic lymph node dissection accounted for 24 interventions, while 1 was a transorbital and transnasal approach to the nasopharynx and the anterior skull base. Cases treated and procedures performed to identify the usability, safety, and effectiveness of the Versius robotic system are summarized in Table 3.

#### 3.2.1. Usability of the Versius Robotic System in the Preclinical Setting

To date, surgeons and fellows engaged in upper GI tract and colorectal surgery, obstetrics/gynecology, urology, and otorhinolaryngology have used the Versius system in a preclinical setting on cadavers and in porcine models [13,22,23,25,26,27]. In an early study by Hares et al. the positive responses of surgeons concerning the performance of the Versius system in proof-of-concept cadaver studies showed that the system could be used successfully in minimal access surgery (MAS) [13]. After the introduction of the Versius system in the market, its usability in MAS was evaluated in simulated clinical settings [22,25]. In a study conducted by Thomas et al. different surgeons and operating team personnel employed the system and successfully completed the respective procedures [25]. The results of the studies showed that the system could be operated proficiently by healthcare professionals trained in laparoscopy as well as those trained in robot-assisted surgery after they had undergone the Versius training program [22,25]. In line with these data, the training program of the Versius system in Butterworth and coworkers’ study was effective; the participants were able to successfully operate the system after they had completed the program, and more surgeons achieved intermediate- and expert-level GEARS scores in validation compared with the first assessment [23]. The first preclinical assessment of the Versius surgical robotic system for transanal total mesorectal excision (taTME) revealed that the ability to work simultaneously bears the theoretical advantage of shortening operating time and thus reducing the overall cost of surgery. It may also allow surgeons to focus on critical parts of the operation by halving the fatigue associated with long, complex procedures such as taTME [28] (Table 2).

#### 3.2.2. Safety and Effectiveness of the Versius System for MAS in Preclinical Studies

The safety and effectiveness of the Versius system have been investigated in preclinical studies [22,25,26,27].

Carey et al. evaluated the utility of the Versius system for gynecologic procedures in a preclinical setting. Several types of gynecological operations were tested. The surgeons evaluated a range of port and BSU positions. With the exception of one case, all procedures were successful. Oviduct removal was also performed safely and effectively in cadaver and porcine models [26]. In the first preclinical assessment of the Versius system for renal and prostate procedures, all procedures (*n* = 24) were completed successfully. One device-related intraoperative complication was noted in a non-recovery pig, showing evidence of thermal injury to the bowel from the monopolar instrument shaft. Only two non-device-related intraoperative complications were recorded (one related to port insertion and one related to replacement of a port that was removed too early). A clinical investigation of the recovery pigs postoperatively revealed no signs of ill health or distress, and all recovery pigs gained weight after surgery [25]. In Haig and coworkers’ study, communication between members of the surgical teams was checked during general surgical procedures and by tasking the surgeon with requesting instrument changes. No related critical task failures were observed for any of the performed tasks, thus validating the safety of these design features. The authors concluded that the Versius system could be used safely by healthcare professionals trained in laparoscopy as well as those trained in robot-assisted surgery [22] (Table 2).

### 3.3. Clinical Safety and Effectiveness of the Versius System in MAS

The results of preclinical studies supported an assessment of the new robot-assisted MAS system in the clinical setting, in the fields of general, gynecologic, and urologic surgery [22,25,26]. This was followed by studies investigating the clinical safety and effectiveness of the Versius system in MAS [20,29,30,31,32,33,34,35]. Investigations in live humans are summarized in Table 4.

Our review comprised a total of 328 patients who had been operated on with this robot system, of whom 61% (197/323) and 49% (126/323) were women and men, respectively. The mean age of the patients was 54.12 years and their average body mass index (BMI), 27.51 kg/m^2^. Based on the reviewed studies, we identified the three categories of colorectal, visceral, and gynecologic surgery. Patient characteristics are shown in Table 5.

#### 3.3.1. Effectiveness of the Versius system in Colorectal Surgery

Of 156 patients who underwent colorectal surgery in five studies [29,30,31,32,34], 62.6% (98/156) were men. The mean age of the patients was 59.86 years. The 135 colorectal surgeries were mainly performed for malignant indications (86.53%). The procedures included colorectal resection, abdominoperineal excision of the rectum, sigmoid colectomy, pan-proctocolectomy, and colostomy formation. The most frequent procedure was right hemicolectomy (30.76%). Patient characteristics are summarized in Table 5.

Based on the reviewed studies, 4% to 6.25% of the surgeries were converted to open procedures [29,30,31]. Postoperative complications varied between 7.4% and 39% [30,31,34]. Readmission and reoperation rates in colorectal surgery using the Versius system were 0–8.8% and 0–9%, respectively [30,31]. Perioperative and postoperative outcomes of patients who underwent colorectal surgery using the Versius system are summarized in Table 6.

#### 3.3.2. Effectiveness of the Versius System in Visceral Surgery

Forty-five patients underwent visceral surgery in two studies [30,33], of whom 60.9% (28/46) and 39.1% (18/46) were men and women, respectively. The mean age of the patients was 59.83 years. All visceral surgeries were performed for benign indications. The procedures included cholecystectomy, appendectomy, inguinal hernia repair, and other types of hernia repair. Patient characteristics are summarized in Table 5.

In the investigation reported by Dixon et al. the majority of patients were day cases. One patient was hospitalized due to urinary retention and pain. No major complications occurred, and no readmissions or reoperations were observed over 30 days [30]. In a study by Kelkar et al. no procedure required conversion to conventional laparoscopic surgery (CLS) or open surgery. All procedures were completed successfully. Intraoperative blood loss was considered negligible (<5 mL) for 7/14 (50%) procedures or minimal (<500 mL) for 7/15 (50%) procedures; two patients were readmitted due to acute gastroenteritis (Clavien-Dindo grade I); the complications were not related to the surgical device. Readmittance rates at 30 and 90 days post-surgery were 1/30 (3.3%) and 2/30 (6.7%), respectively [33].

### 3.4. Effectiveness of the Versius System in Gynecologic Surgery

The first clinical studies on the use of the Versius system in gynecologic surgery [30,33] and a report on 45 robotic radical hysterectomies were published recently in India [20,35]. One hundred and six women who underwent gynecologic surgery in four studies were included [20,30,33] in the review. The mean age of the patients was 47.1 years and their average BMI 27.56 (kg/m^2^). Fifty-five gynecologic surgeries were performed for malignant indications (43.8%). The procedures included total robotic hysterectomy, salpingo-oophorectomy, ovarian cystectomy, diagnostic laparoscopy, oophorectomy, and fallopian tube recanalization procedures. The most frequent procedure was robotic hysterectomy (52.1%). Patient characteristics are summarized in Table 5.

Dixon et al. reported no conversion from robot-assisted surgery to other procedures, no reoperations, and no major complications. Three patients were readmitted within 30 days (for small bowel obstruction, pain, and intra-abdominal collection), and all were managed conservatively [30]. In a report published by Puntambekar et al. 12 and 18 of 30 patients had endometrial and early cervical cancer (IA2-4, IB2-10, and IIA1-4), respectively. The total operating time from docking to the removal of ports was 104 min (60–150 min), and the average blood loss was 60 mL (50–100 mL). The urinary catheter was removed by the 7th day in the majority of patients. No patient experienced urinary retention. A type II/B RRH was performed in patients with cervical cancer, stage IA1 and IA2. Patients with stage IB and IIA disease underwent type III/C1 RRH. All patients with advanced endometrial cancer underwent type C1 RRH. Postoperative complications were seen in two patients. Both injuries were ureterovaginal fistulae on the 8th day post-surgery. One patient was managed with a double-J stent while the other underwent laparoscopic ureteral reimplantation. No patient required prolonged hospitalization and no death was registered within 30 days post-surgery [20].

In a study conducted by Kelkar et al. no procedure required conversion to CLS or open surgery, all procedures were completed successfully, and the authors registered no intraoperative complications. Intraoperative blood loss was considered negligible (<5 mL) for 12/16 (75%) procedures or minimal (<500 mL) for 4/30 (25%) procedures; only one case (3.3%) required the use of blood transfusion products. No patient was returned to the OR within or after 24 h post-surgery, and no readmission was registered at 30 and 90 days [33]. In another investigation conducted by Kelkar et al. 15 total laparoscopic hysterectomies (TLH) were performed using the Versius system, and 1 procedure was converted to open surgery due to the patient’s elevated BMI. All other procedures were completed as planned and no complications were registered. The operating time ranged between 110 min and 345 min (median 205 min), and the estimated blood loss for all patients was <500 mL. No adverse events were reported at the 30-day follow-up [35].

## 4. Discussion

Robot-assisted surgery is still rather new, but it is a cutting-edge development in surgery with far-reaching implications. While improving precision and dexterity, this technology allows surgeons to perform operations that were traditionally not amenable to minimal access techniques [41]. The last 20 years have witnessed the emergence of several robot-assisted surgical devices with the purpose of overcoming some of the challenges associated with MAS [13]. Since previous robot-assisted surgical devices had some limitations, CMR developed and designed Versius, a teleoperated robotic surgical system, to assist surgeons in performing MAS and overcoming the challenges of the currently existing systems [13,21]. Versius was designed to enhance team communication, improve the surgeon’s work environment, and prolong the surgeon’s career as a result of improved ergonomics; these variables had been identified as major problems in the use of previous robotic surgical systems [13]. First reports on the use of the Versius system in MAS were published one year after the introduction of the device in the market [13,22,23,24,25,26,27,28,42].

As the Versius system is still quite new, we assessed the feasibility of the device in colorectal, visceral, and gynecologic surgery. Sixteen studies were included in the present systematic review. To date, 100 MAS procedures in human cadavers and live animals have been performed by 68 surgical teams [13,22,23,24,25,26,27,28]. Studies showed that the training program for the Versius system is effective. The system can be used proficiently by healthcare professionals trained in laparoscopy as well as those trained in robot-assisted surgery [13,22,23,25]. The large majority of the procedures using the Versius system were successful [22,26,27,29]. No device-related critical task failures were encountered. The studies confirmed that Versius can be used easily and safely [22,25,26,27], and MAS can be performed successfully by specialists in their respective fields [13,28,29,30,31]. The ability to work simultaneously reduced operating times as well as overall surgical costs. It may also allow surgeons to focus on critical parts of the operation by halving the fatigue associated with long and complex procedures [28].

The clinical effectiveness of the Versius system in MAS was investigated in eight studies conducted in 2020 and 2021 [20,29,30,31,32,33,34,35]. A total of 328 patients were operated on with this robot system, of whom 48.3%, 14.2%, and 37.5% were colorectal, visceral, and gynecological cases, respectively [29,30,31,32,34]. In colorectal surgeries, postoperative and major complications within 30 days were 7.4–39% [30,31,34] and 0–9%, respectively [30,31]. In gynecologic surgery, no adverse events were reported at the 30-day follow-up [35]. No major complications and no readmissions or reoperations were registered in visceral and gynecologic surgery [20,30,33,35]. Readmission and reoperation rates in colorectal surgery were 0–9% [30,31]. No procedure required conversion to CLS or open surgery, and all procedures were completed successfully [30,33,35]. Based on the studies reviewed in the current report, this robot can be used safely and effectively in MAS [20,29,30,31,32,33,34,35].

In 1797 operations performed with the da Vinci system, Kim et al. noted conversions to open or laparoscopic surgery in 0.17% of cases (3/1797) [43]. Siaulys and coworkers performed 100 gynecological robotic surgeries in Klaipeda, Lithuania, using the Senhance robotic platform. The mean duration of the patients’ hospital stay was 4 ± 2.3 days, range 1–14 days. Six (6%) conversions were reported: one to laparoscopy and five to open surgery [44]. Two intraoperative complications were observed, and one patient was readmitted in the early postoperative period due to severe vaginal bleeding [45]. Three complications occurred during 30 days after the operation [44]. The authors concluded that the Senhance system is safe and feasible for use in MAS [44,45,46].

In view of differences in surgical procedure, stages of cancer, and the surgeons performing the operations, no final conclusions can be drawn about the robotic procedures reviewed above. It would be appropriate to design a randomized controlled trial (RCT) for different types of surgery (colorectal, visceral, urology or nephrology, and gynecological procedures) across different surgical platforms to determine the advantages and disadvantages, and to overcome the limitations of a master-slave system.

Surgeons have always welcomed technological advances that would benefit their patients. Some of the previous robot systems had limitations such as loss of triangulation, instrument clashing, and limited assistant workspace. New robot systems, including the Versius system, were especially designed to overcome these challenges. Despite the increasing use of this platform in surgery, the body of published literature on the advantages and disadvantages of the system is still scarce. We identified a mere eight clinical studies, and the majority of these comprised small numbers of samples. However, the promising early results of the studies confirmed the safety and efficacy of the Versius system in MAS. It would appear that the Versius system is a developing technology rather than one set to immediately replace CLS. The Versius system may offer yet undisclosed advantages. The ergonomic features of the device could enhance the surgeon’s skills by reducing physical and mental strain during the intervention.

Further studies will pave the way for a global learning platform. The latter will permit the first version of Versius to develop into more advanced forms such as 2.0 or 3.0. CMR Surgical has broached an exemplary academic and scientific pathway of commercial access for Versius. Accompanying studies, re-evaluation processes, and a learning program based on the exchange of communication with users and trainers will enhance the uses and benefits of the Versius robot system.

Following construction of the device, a number of preclinical studies and feasibility investigations were performed before the robot was employed in living patients. The majority of living patients operated on by the use of the device have been involved in clinical studies. This strengthens our confidence in being able to advance the technology of the device and improve the quality of treatment for patients.

CO_2_ emissions of robotically assisted surgeries, considering both direct and indirect factors, have a significant impact on the environment. Therefore, it will be necessary to compare different platforms in terms of their carbon footprints and determine their respective environmental effects. Clinicians, administrators, and policy-makers will then be able to adopt appropriate sustainable measures.

Future research and development should be focused on wider applications, improving outcomes, increasing availability, and reducing costs.

### Strengths and Limitations

In a first systematic review, we assessed the feasibility and safety of the Versius system in MAS. The limitations of the studies reviewed in this systematic report include their scope, rather small sample sizes, short follow-up periods, and quality. No randomized controlled trials (RCTs) have been performed yet on the use of this robot system in MAS. We recommend future RCTs to evaluate the efficacy of the Versius system, long-term outcomes, patients’ quality of life, and the cost of this surgical robot system in MAS. Direct comparisons of robotic platforms (DVSS, REVO-I, Senhance, etc.) will be necessary to assess clinical outcomes, their potential advantages and disadvantages, surgeon preferences, and the economic and environmental sustainability of the devices.

## 5. Conclusions

Versius, albeit a new platform, is reshaping robot-assisted surgery across all disciplines. Early preclinical and clinical results have confirmed the potential ability of this system to reform MAS robot-assisted surgery. However, the data analyzed in the present review should be viewed with caution pending the availability of data from randomized clinical trials. Future studies focusing on oncologic indications must include the crucial outcome of patient survival.

## Figures and Tables

**Figure 1 jcm-11-03754-f001:**
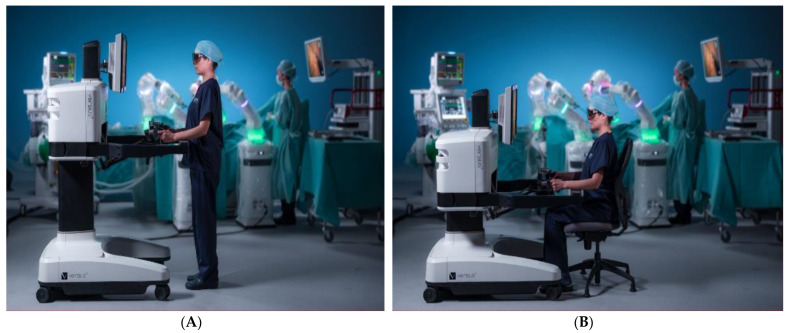
Versius surgical robotic system: (**A**) the surgeon at the console in standing and (**B**) sitting position.

**Figure 2 jcm-11-03754-f002:**
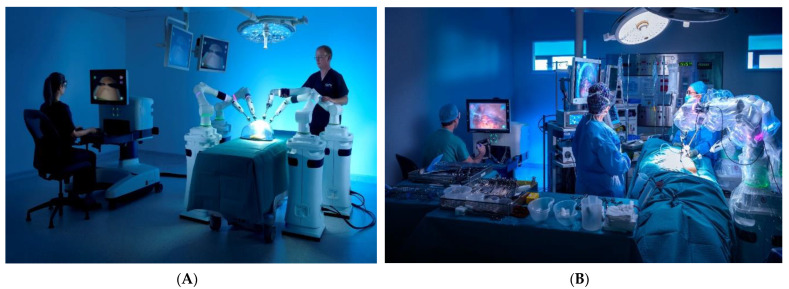
(**A**) Overview of the Versius surgical robotic system in a training setup (**B**) Versius surgical robotic system in OR use.

**Figure 3 jcm-11-03754-f003:**
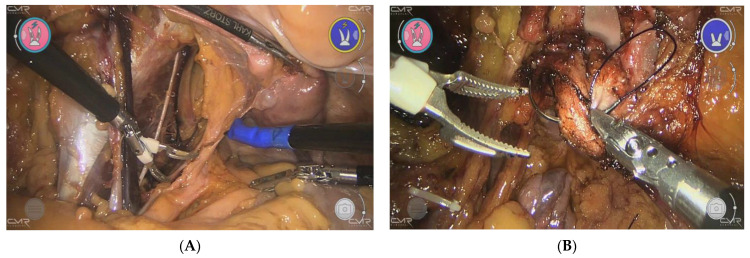
Endoscopic view of the Versius system. (**A**) Posterior exenteration after systematic pelvic lymphadenectomy. (**B**) Suturing of the vaginal cuff using a monofilament suture. Needle holder in the right hand and bipolar clamp in the left hand.

**Table 1 jcm-11-03754-t001:** Characteristics of the Versius robotic surgical platform (multi-port robotic system) [21].

Manufacturer	CMR Surgical
**Robotic platform**	Versius
**Country of origin**	United Kingdom
**Approach**	Laparoscopic
**Approval/year**	European CE Mark (2019), TGA approval (2020)
**Patient cart**	Multiple
**Surgeon console**	Open/3D glasses
**Number of consoles**	2 possible
**Arm configuration**	Modular
**Number of arms**	7 (experimental only)
**Camera diameter**	10 mm
**Instrument diameter/DOF**	5 mm/7°
**Ability to employ trocars**	Yes
**Foot pedal control**	No
**Gravity compensation**	Yes
**Ability to operate in two fields**	Yes (included in the package)
**Effector arm lifespan**	Abdominal and thoracic surgery
**Simulator available**	Yes
**Fields of application**	Abdominal surgery
**Advantages**	- Easy access due to flexible positioning of the robotic working port. - The system provides haptic feedback and standard re-usable instruments for reducing costs.
**Disadvantages**	- Application only in a few hospitals and only in abdominal surgery so far. - No distinct surgical instruments for head and neck surgery, no adaptations to the narrow surgical field in the head and neck.

**Table 2 jcm-11-03754-t002:** Feasibility of the Versius system for healthcare professionals.

Author/Year	Title	Summary of Method	Country	Main Results	Quality of Article
**Kayser/2022** [40]	Evaluation of the Versius Robotic Surgical System for Procedures in Small Cavities	**Study design:** Unknown **Sample size:** Eight cardboard boxes **Surgical team:** Seven members (a gynecologic surgeon for adults, experienced in the use of Versius, and six vascular surgeons, pediatricians, and general practitioners. inexperienced in the use of Versius) **Program:** The procedures, two single stitches with two square knots each, were performed in every box.	Germany	All procedures were performed successfully in all boxes	High risk of bias
**Butterworth/2021** [23]	Assessment of the training program for Versius, a new innovative robotic system for use in minimal access surgery	**Study design:** Pilot study **Sample size:** Seventeen surgical teams **Audience group:** The surgeons worked in the following specialties: upper GI tract (*n* = 5), colorectal (*n* = 4), obstetrics/gynecology (*n* = 4), and urology (*n* = 4). **Training program:** The on-site training program comprised a platform overview and basic training (day 1), training implementation (day 2), training consolidation (day 3), and training reinforcement (day 4 [half-day]). **Length of program:** A 3.5-day program following 10 h of online didactic training.	USA	The participants showed an overall improvement of their performance during the study, with a mean Global Evaluative Assessment of Robotic Skills Score (GEARS) of 21.0 ±1.9 in Assessment 1. The score increased to 23.4 ± 2.9 in Validation. The greatest improvements were seen in the domains of depth perception and robotic control. The greatest differences were observed when stratifying by robotic experience. Those with extensive experience consistently scored higher than those with some or no experience.	High risk of bias
**Morton/2021** [24]	Preclinical evaluation of the versius surgical system, a new robot-assisted surgical device for use in minimal access general and colorectal procedures	**Study design:** Clinical trial **Sample size:** Thirty-eight procedures in human cadavers and 11 procedures in pigs. **Type of surgery:** Nine types of general and colorectal procedures were performed in cadavers across the following anatomical regions: right and left hypochondrium, epigastrium, and right and left iliac fossae. Cholecystectomy (*n* = 6) and small bowel enterotomy (*n* = 5) procedures were performed in pigs.	UK & USA	Of 38 procedures, 35 (92.1%) were completed successfully; two procedures could not be completed due to unsuitable port placement, and one due to the physical condition of the cadaver. The port and BSU positions permitted good surgical access and reach; surgical access and reach were given a median score of 6 or more on the visual analog scale (VAS) for seven of eight procedures.	High risk of bias
**Thomas/2021** [25]	Preclinical Evaluation of the Versius Surgical System, a New Robot-assisted Surgical Device for Use in Minimal Access Renal and Prostate Surgery	**Study design:** Pilot study **Sample size:** Twenty-four procedures were completed successfully in cadavers by eight different lead surgeons. **Type of surgery:** Radical nephrectomy, prostatectomy, and pelvic lymph node dissection. **Surgical team:** Experienced renal and prostate surgeons. **Program:** Cadaver sessions were conducted to evaluate the ability of the system to complete all surgical steps required for a radical nephrectomy, prostatectomy, and pelvic lymph node dissection. A live animal (porcine) model was also used to assess the surgical device in performing radical nephrectomy safely and effectively. Surgical access and reach were evaluated by the lead surgeon on a visual analog scale.	UK	Positioning of the ports and bedside units reflected the lead surgeon’s preferred laparoscopic set-up and enabled good surgical access and reach, as quantified by a median visual analog score of ≥6.5. All radical nephrectomies performed in pigs were completed successfully, with no device- or non–device-related intraoperative complications.	High risk of bias
**Carey/** **2020** [26]	Preclinical evaluation of a new robot-assisted surgical system for use in gynecology minimal access surgery	**Study design:** Observational study with cadaver and live animal surgery components or preclinical study. **Sample size:** A variety of gynecologic procedures were performed on 11 female cadavers with no previous abdominal or pelvic surgery. The cadavers encompassed a wide spectrum of BMIs, reflecting the sizes and shapes of actual human anatomy. Six oviduct removals (non-recovery *n* = 2, recovery *n* = 4) were performed in pigs as a surrogate for robot-assisted total laparoscopic hysterectomy (RALH). **Surgical team:** The four surgeons who performed the procedures on cadavers were accredited and practicing high-volume gynecologic surgeons, performing > 50 complex laparoscopic procedures/year. **Program:** Cadaveric sessions were conducted to evaluate the ability of the system to complete all surgical steps required for a robot-assisted total laparoscopic hysterectomy. A live animal (porcine) model was used to assess the system in performing oviduct removal as a surrogate for robot-assisted total laparoscopic hysterectomy.	UK	**Procedure completion in cadavers:** In total, 16/17 procedures were completed successfully. Positioning of the ports and bedside units reflected the surgeons’ preferred laparoscopic setup and enabled good surgical access and reach, as exemplified by the high procedure completion rate. Oviduct removal procedures performed in pigs were all completed successfully by a single surgeon. **Safety in live animals:** All procedures were completed successfully. One device-related intraoperative complication was noted in a non-recovery pig, with evidence of thermal injury to the bowel from the monopolar instrument shaft. Only two non-device-related intra-operative complications were recorded (one related to port insertion and one related to replacement of a port that was removed too early). Clinical observation of the recovery pigs postoperatively revealed no signs of ill health or distress, and all recovery pigs gained weight post-surgery. The macroscopic post-mortem examination revealed minor signs of inflammation around one port site, and cysts on the top of the vaginal cuff in two of four pigs. Removal of the cyst in one pig disclosed an open vaginal cuff. In all pigs, the surrounding organs appeared healthy and had no signs of injury or inflammation.	Low risk of bias
**Faulkner/2020** [27]	Combined robotic transorbital and transnasal approach to the nasopharynx and anterior skull base: Feasibility study	**Study design:** unknown **Sample size:** one **Surgical team:** One surgeon performed procedures in the skull base and the nasopharynx on cadavers.	UK	The study showed that a combined robotic approach to the skull base and the nasopharynx is feasible. Combined transnasal and transorbital ports overcome the funnel effect, allowing current robotic instruments to operate within this space with a limited risk of collision.	High risk of bias
**Haig/2020** [22]	Usability assessment of Versius, a new robot-assisted surgical device for use in minimal access surgery	**Study design:** Pilot study **Sample size:** Seventeen surgical teams participated in the study. **Audience group:** A lead surgeon, an assistant surgeon, a scrub nurse, and a circulating nurse. Upper gastrointestinal (GI) (*n* = 5), obstetrics and gynecology (OB/GYN) (*n* = 4), urology (*n* = 4), and colorectal (*n* = 4). **Training program:** Transport and storage at the hospital, generic surgical procedure, post procedure. **Length of program:** A 3.5-day program following 10 h of online didactic training.	USA	Seventeen surgical teams participated in the study and all were experienced in laparoscopic surgery. Surgical teams performed 11,633 tasks. Of these, 7501 were critical for safe and effective use of Versius, while 4132 were non-critical. No critical task failures were observed. Of all completed tasks, 98% were recorded as a pass or a pass with difficulty.	High risk of bias
**Atallah/2019** [28]	Assessment of the Versius surgical robotic system for dual-field synchronous transanal total mesorectal excision (taTME) in a preclinical model: will tomorrow’s surgical robots promise newfound options?	**Study design:** Pilot study **Sample size:** Three surgeons **Audience group:** Fellowship-trained colorectal surgeons with extensive experience in laparoscopic surgery and the da Vinci surgical system. **Training program:** After a dry laboratory introduction of the Versius system, each surgeon was given the opportunity to advance his/her training on a cadaveric model. Each surgeon performed either splenic flexure mobilization, sigmoid colectomy, or taTME. **Length of program:** 2 days.	USA	Using the modular robotic system, one surgeon performed the abdominal portion of the operation, including colonic mobilization and vascular pedicle ligation. A second surgeon simultaneously performed the transanal portion of the operation to the point of rendezvous at the peritoneal reflection, where the operation was completed cooperatively. The operation was successfully completed in 195 min, demonstrating the preclinical feasibility of this unique approach with an emerging robotic system.	High risk of bias
**Hares/2019** [13]	Using end-user feedback to optimize the design of the Versius Surgical System, a new robot-assisted device for use in minimal access surgery	**Study design:** Pilot study **Sample size/Audience group/Training program:** **Formative arm study 1:** Ten scrub nurses/OR technicians; **Arm usability study 2:** Seven members of the scrub team; **Surgeon handgrips formative study:** Eight surgeons with varying levels of experience in laparoscopic and robotic surgery; **Grips study 2 and Instrument tip exploratory study:** Ten robotic surgeons; **Console usability study:** Thirteen surgeons from a range of hospitals and with various levels of experience; **Surgeon console study:** Eight laparoscopic/robotic surgeons with and without robotic experience; **Workflow study 1:** Four different surgical teams; **Workflow study 2:** Four different surgical teams (2× urology, 1× colorectal, 1× gynecology).	UK	Feedback led to the development of a novel mobile design with independent arm carts and surgical console, linked by supported serial or parallel connections, providing maximum flexibility in the OR. Instrument tips were developed on the basis of the surgeons’ preferences and wristed at the tip, providing seven degrees of freedom within the patient. Multiple handgrip designs were assessed by surgeons. Of these, a ‘game controller’ design was rated most popular and usable. An open surgical console design allowing multiple working positions was rated highest by surgeons and surgical teams.	High risk of bias

**Table 3 jcm-11-03754-t003:** Cases/procedures performed in studies.

Cases/Procedures	Human Cadavers	Total	Live Animals	Total
Cholecystectomy (*n* = 17), antegrade dissection of the gallbladder (*n* = 1), distal pancreatectomy with splenectomy (*n* = 1), Nissen fundoplication (*n* = 1), splenectomy (*n* = 2), splenic flexure mobilization (*n* = 3), left hemicolectomy combined with low anterior resection (*n* = 5), low anterior resection (*n* = 5), total mesorectal excision (*n* = 5), splenic flexure mobilization (*n* = 1), sigmoid colectomy (*n* = 1), or taTME (*n* = 1).		41	Cholecystectomy (*n* = 6) and small bowel enterotomy (*n* = 5)	11
Radical nephrectomy (transperitoneal and retroperitonal) ( *n* = 16), prostatectomy (transperitoneal) (*n* = 3), Retzius-sparing prostatectomy (*n* = 1), pelvic lymph node dissection (*n* = 4).		24		
Burch colposuspension (*n* = 3), paravaginal repair of the vaginal wall (*n* = 1), sacrocolpopexy (*n* = 1), sacrohysteropexy (*n* = 3), subtotal laparoscopic hysterectomy with sacrocervicopexy (*n* = 3), robot-assisted total laparoscopic hysterectomy (RALH) (*n* = 6).		17	Oviduct removals (*n* = 6)	6
One surgeon performed procedures in the skull base and the nasopharynx on cadavers.		1		
Total		83		17

**Table 4 jcm-11-03754-t004:** Clinical effectiveness of the Versius system in minimal access surgery.

Author/Year	Title	Summary of Method	Country	Results	Quality of Article
**Collins/** **2021** [29]	Implementation of the Versius robotic surgical system for colorectal cancer surgery: First clinical experience	**Study design:** Prospective series **Sample size:** 32 patients (men; *n* = 15, women; *n* = 17). **Type of surgery:** Right hemicolectomies (*n* = 18) and anterior resections (*n* = 14).	UK	Estimated blood loss was 150 mL; range < 100 to <500 mL. Eight patients experienced grade II morbidities (22%); no serious morbidities and no mortalities were observed. The mean period of time until recovery of bowel function was 2.9 days (1–6 days). The average duration of the hospital stay was 5.3 days; median 4 days (range 2–20 days). All resections were R0; the average lymph node yield was 20 nodes (8–46 nodes). The results confirmed the safety of Versius and its feasibility for colorectal resection.	Good
**Dixon/** **2021** [31]	Major colorectal resection is feasible using a new robotic surgical platform: the first report of a case series	**Study design:** Case series Sample size: 23 operations. **Type of surgery:** Left- (*n* = 14) and right- (*n* = 9) sided colon resections.	UK	Fifty-seven percent of the patients were male; a malignant indication for surgery was present in 70% of cases. Only one operation (4%) was converted from the robotic to the open approach. The median length of the postoperative stay was 5 days; no readmissions were observed within 30 days. The study showed that the Versius system is feasible for use in major colorectal resection. These early results from a robot-naïve center are promising. They indicate an expanding robotic market and highlight the need for further evaluation.	Good
**Dixon/** **2021** [30]	Initiation and feasibility of a multi-specialty minimally invasive surgical programme using a novel robotic system: A case series	**Study design:** Case series **Sample size:** 160 patients. Type of surgery: Colorectal (*n* = 68), gynecology (*n* = 60), and general surgery (*n* = 32).	UK	The conversion rate to open surgery in gynecology was 0%. The median length of the hospital stay for gynecologic surgery was 1 day. The conversion rate to the open procedure in colorectal surgery was 4.4%. The median duration of the hospital stay for colorectal surgery was 6 days. The Versius system is safe and feasible for use in a multi-specialty minimally invasive surgery program, including colorectal, general surgical, and gynecological cases. The operative volume can be safely and easily scaled up in a district general hospital setting without prior robotic surgical experience.	Good
**Huddy/2021** [32]	Experiences of a “COVID protected” robotic surgical center for colorectal and urological cancer in the COVID-19 pandemic	**Study design:** Unknown Sample size: 2. **Type of surgery:** Sigmoid colectomy (*n* = 1) High anterior resection (*n* = 1).	UK	Both Versius cases (one sigmoid cancer and one upper rectal cancer) were discharged on day 2 without stomas and with no postoperative complications.	Fair
**Kelkar/2021** [33]	Interim safety analysis of the first-in-human clinical trial of the Versius surgical system, a new robot-assisted device for use in minimal access surgery	**Study design:** Clinical trial **Sample size:** 30 patients. **Type of surgery:** Elective minor or intermediate gynecological (*n* = 16) or general surgical procedures (*n* = 14).	India	All procedures were completed successfully without the need for conversion to minimal access surgery (MAS) or open surgery. No patient returned to the OR within 24 h after surgery; readmission rates at 30 and 90 days postsurgery were 1/30 (3.3%) and 2/30 (6.7%), respectively. This first-in-human interim safety analysis demonstrates that the Versius surgical system is safe and can be used successfully to perform minor or intermediate gynecological and general surgery procedures.	Good
**Puntambekar/2021** [34]	Colorectal cancer surgery: by Cambridge Medical Robotics Versius Surgical Robot System—a single-institution study. Our experience	**Study design:** Unknown **Sample size:** 31 patients, 23 men and 8 women. **Type of surgery:** Colorectal adenocarcinoma.	India	Mean age 55.6 years. The mean robotic operating time was 51 min. The mean robot docking and undocking time was 17 and 5 min, respectively. The mean duration of the hospital stay was 7 days. Longitudinal and circumferential resection margins were negative in all patients. Histopathological reports for 27 of 31 patients showed complete total mesorectal excision (TME). The advantages of the Versius robot include dexterity, clarity of vision, intuitive movements, and the potential to translate these technical features into oncological safety.	Good
**Puntambekar/2020** [20]	Feasibility of robotic radical hysterectomy (RRH) with a new robotic system. Experience at Galaxy Care Laparoscopy Institute	**Study design:** Clinical trial **Sample size:** 30 patients with early cervical cancer/endometrial cancer. **Type of surgery:** Radical hysterectomy.	India	The mean operating time was 104 min, and the mean total lymph node yield 24.7. The average blood loss was 60 mL. The average length of the hospital stay was 2.1 days, and the majority of patients were catheter free by 1 week. Two patients developed uretero-vaginal fistulae on the 8th day of surgery. The study demonstrated the feasibility, safety, and efficacy of RRH using the Versius robotic system.	Good
**Kelkar/2020** [35]	First-in-human clinical trial of a new robot-assisted surgical system for total laparoscopic hysterectomy	**Study design:** Clinical trial **Sample size:** 15 patients with adenomyosis, abnormal uterine bleeding, uterine fibrosis, endometriosis, and menorrhagia. **Type of surgery:** Total laparoscopic hysterectomy (TLH).	India	One procedure was converted to open surgery due to the patient’s elevated BMI. All other procedures were completed as planned, with no recorded complication.	Fair

**Table 5 jcm-11-03754-t005:** Characteristics of operated patients.

Characteristics	Colorectal (*n* = 156) [29,30,31,32,34]	Visceral (*n* = 46) [30,33]	Gynecology (*n* = 121) [20,30,33,35]	Total (*n* = 323)
**Gender, male/female**	98/58	28/18	0/121	126/197
**Age (years), mean**	59.86	50.83	47.1	54.12
**Body mass index (kg/m^2^), mean**	27.36	27.86	27.56	27.51
**Indication for surgery** **malignant/benign**	135/21	0/46	53/65	188/135
**Case/procedure**	Colorectal resection (right hemicolectomies (right/extended hemicolectomies and ileocolic resection (***n* = 48**)), left hemicolectomies (***n* = 17**), anterior and high anterior resection (***n* = 40**), low and ultra-anterior resection (***n* = 31**)), abdominoperineal excision of the rectum (***n* = 10**), sigmoid colectomy and upper rectal cancer (***n* = 3**), pan-proctocolectomy (***n* = 1**), completion proctectomy for inflammatory bowel disease (one with an ileoanal pouch formation) (***n* = 2**), stoma reversals (two Hartmann’s colostomy reversals and one intracoporeal ileocolic re-anastomosis) (***n* = 3**), colostomy formation (***n* = 1**).	Cholecystectomy (***n* = 16**), appendectomy (***n* = 4**), inguinal hernia repair (one bilateral) (***n* = 18**), and other hernia repair (including ventral, incisional, and parastomal hernias) (***n* = 8**).	Total robotic hysterectomy and bilateral salpingo-oophorectomy (***n* = 63**), robot-assisted total laparoscopic hysterectomies (***n* = 36**), bilateral salpingo-oophorectomies (***n* = 6**), ovarian cystectomy (***n* = 5**), unilateral salpingo-oophorectomy (one involving extensive adhesiolysis) (***n* = 3**), diagnostic laparoscopy case (***n* = 5**), oophorectomy (***n* = 2**), fallopian tube recanalization procedure (***n* = 2**).	

**Table 6 jcm-11-03754-t006:** Perioperative and postoperative outcomes of patients who underwent colorectal surgery using the Versius system *.

		Collins (*n* = 32) [29]	Dixon (*n* = 23) [31]	Dixon (*n* = 68) [30]	Huddy (*n* = 2) [32]	Puntambekar (*n* = 30) [34]
**Console time (minutes) (range)**		Right hemicolectomy 111 (64–213) Anterior resection 204 (85–242)	166 (range 75–320)	159 (range 21–320)	-	51 (43–80)
**BSU set-up time (minutes), median (range)**		-	17 (7–39)	11 (5–39)	-	-
**Conversion to open, *n* (%)**		Two cases (6.25%)	One operation (4%)	3 (4.4%)	0	0
**Pain score (out of 10),** **median (range)**	Day 1 postop	-	4 (0–8)	4 (0–10)		
	Day 2 postop	-	4.5 (0–10)	5 (0–10)		
	Day 3 postop	-	4 (0–8)	4 (0–10)		
**Intraoperative complications**		Four patients (12.5%). Covering loop ileostomy either due to a very low anastomosis or due to preoperative radiotherapy.	-	-	-	21 patients (70%) Bleeding (*n* = 2), serosal tear in the bowel (*n* = 1), loop ileostomy (*n* = 18)
**Postoperative complications**			Nine patients (39%). One patient developed a postoperative ileus, and three patients received antibiotics for superficial wound infections. Two patients developed postoperative urinary retention requiring re- catheterization. One patient had a urinary tract infection and one developed a pulmonary embolism which was treated with anticoagulation.	Five patients (7.4%). Intra-abdominal collection requiring a radiological drain (*n* = 1), anastomotic leak necessitating return to the operating room and the creation of an end colostomy (*n* =1), extraction site hernia needing open surgical repair (*n* =1), hematoma requiring washout in the operating room (*n* = 1), perineal wound infection needing EUA and a vacuum dressing (*n* =1).	0	Six patients (20%), Bowel obstruction (*n* = 2), surgical site infection (*n* = 3), anastomotic dehiscence (*n* = 1)
**Mean period of time to recovery of bowel function in days (range)**		2.9 (1– 6).	-	-	-	3 (2–4)
**Duration of hospital stay in days (range)**		4 (2–20) †	5 (range 3–34) †	† 6 (3–34)	2 †	6 (5–12) **
**Major complications within 30 days, *n* (%)**		-	2 (9%)	5 (7.4%)	0	-
**Readmission within 30 days, *n* (%)**		-	0	6 (8.8%) Three patients were readmitted with intra-abdominal collection, one with a wound infection and two with postoperative vomiting.	0	-
**Reoperation within 30 days, *n* (%)**		-	2 (9%)	4 (5.9%)	0	-

Abbreviations: BSU = bedside unit, EUA = examination under anesthesia; * All studies were published in 2021; † Median; ** Mean.

## Data Availability

The data presented in this study are available on request from corresponding author.

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
