# Peer review of "Assessment of the Versius Robotic Surgical System in Minimal Access Surgery: A Systematic Review"

_jcm, 2022, doi:10.3390/jcm11133754_

Round 1
Reviewer 1 Report
Thanks for the opportunity to review a nice paper which shows the versatility of the Versius system whilst also touching upon the limitations and potential areas of need for improvement in future.
Small recommendations and suggestions
1) this is a scoping review and not a sysematic review
1)hope appropriate permissions were sought go use figure 1
2) In introduction. A mention of ergonomics will be useful comparing Versius vis a vis davinci
3) The carbon footprint of using Versius may be touched upon
4) OT space footprint can also be quoted as its a smaller robotic system
Another reference that may be useul is
Morris B. Robotic surgery: applications, limitations, and impact on surgical education. MedGenMed. 2005;7(3):72. Published 2005 Sep 27.
In discussion: Line 332 -337 regarding mechanical failure rate seem bit abrupt and out of place.maybe omitted or concised
The statement on having a RCT bn 3 surgical systems (361-363) : yes its valid but hypothetical to quote. Instead suggesting a appropriate comparative trial for a particular surgery eg: robotic prostatectomy across different surgical platforms to outline advantages and limitations will help overcome and improve shortcomings of a master- slave system.
Line 364 is grammaticallly incoreect and not needed
The weak point of this study is the way conclusions are writen:
Rephrase them maybe as :
Versius albeit a new platform is reshaping the way robotic surgery is performed across all disciplines of surgery. Early pre clinical and clinical results reaffirm.that this system will reform MAS robotic surgery as it gains prominence across all disciplines.
It's potential to bring down costs of robotic surgery will be a added bonus , however this needs further clinical studies
Author Response
|
Reviewer 1 |
|
|
Thanks for the opportunity to review a nice paper which shows the versatility of the Versius system whilst also touching upon the limitations and potential areas of need for improvement in future. |
|
|
1) this is a scoping review and not a sysematic review |
Thank you for this valuable comment. The distinction between systematic and scoping is based on: 1. having or creating hypotheses (we were going to test the hypotheses) 2. having goals or objectives (in this study we were going to answer to objectives). Since we had the hypotheses before the conducting the study, and we were in search of the answer to our questions, so we did this study according to the systematic review process, then we can call this paper as a systematic review. |
|
1)hope appropriate permissions were sought go use figure 1 |
Yes, those pictures were included by the approval by CMR. |
|
2) In introduction. A mention of ergonomics will be useful comparing Versius vis a vis davinci |
Thank you for this valuable comment. Ergonomics was mentioned in the introduction and discussion sections. Please see highlighted parts. Lines 62-64, 313-316, 369-370. |
|
3) The carbon footprint of using Versius may be touched upon |
We appreciate this valuable comment which we completely didn’t pay attention to. Based on this valuable comment we mentioned to carbon footprint of Versius and the others robots in introduction and discussion. Based on this valuable comment we added a recommendation for future studies. Please see highlighted parts. Lines 40-43, 382-386. |
|
4) OT space footprint can also be quoted as it’s a smaller robotic system |
We appreciate this valuable comment which we completely didn’t pay attention to. Based on this valuable comment we mentioned to carbon footprint of Versius and the others robots in introduction and discussion. Based on this valuable comment we added a recommendation for future studies. Please see highlighted parts Lines 40-43, 382-386. |
|
Another reference that may be useful is Morris B. Robotic surgery: applications, limitations, and impact on surgical education. MedGenMed. 2005;7(3):72. Published 2005 Sep 27. |
Thank you for this valuable comment. Based on this comment we strated the discussion of this paper with use of valuable sentences from the mentioned study. This reference was added (ref:38). Lines: 305-308 |
|
In discussion: Line 332 -337 regarding mechanical failure rate seem bit abrupt and out of place. Maybe omitted or concised |
Thank you, these sentences were omitted. |
|
The statement on having a RCT bn 3 surgical systems (361-363) : yes its valid but hypothetical to quote. Instead suggesting a appropriate comparative trial for a particular surgery eg: robotic prostatectomy across different surgical platforms to outline advantages and limitations will help overcome and improve shortcomings of a master- slave system. |
The sentences were edited based on the suggestion.
Lines: 355-357. |
|
Line 364 is grammaticallly incoreect and not needed |
As regards language, our in-house editor and native speaker has checked the paper thoroughly. She found some typographical errors and made several changes in style and syntax. If the editors still happen to find grammatical errors, we would be grateful if a few of these could be explicitly mentioned, as we could then pass these on to our editor. |
|
The weak point of this study is the way conclusions are writen: Rephrase them maybe as : Versius albeit a new platform is reshaping the way robotic surgery is performed across all disciplines of surgery. Early pre clinical and clinical results reaffirm.that this system will reform MAS robotic surgery as it gains prominence across all disciplines. |
Thank you for this valuable conclusion. Conclusion was replaced by the reviewer suggestion. Line 400-402 |
|
It’s potential to bring down costs of robotic surgery will be a added bonus , however this needs further clinical studies |
Thank you, we try to consider this issue in several parts of the text. Lines 57, 200, 328, 395, Table 1. |

Reviewer 2 Report
Dear Authors,
in this systematic review you describe pros and cons of Versius robotic system. I read this manuscript with great interest and in my opinion it fulfills systematic review guidelines. I would suggest only that subheadings 3.3 and 3.4 should be a little shortened beacuse all date are to found in tables 4-6.
Author Response
|
in this systematic review you describe pros and cons of Versius robotic system. I read this manuscript with great interest and in my opinion, it fulfills systematic review guidelines. I would suggest only that subheading 3.3 and 3.4 should be a little shortened beacuse all date is to found in tables 4-6. |
Thank you for this valuable comments. All duplicate dates in the text and tables were deleted. |
